# Failure Characterization of Al-Zn-Mg Alloy and Its Weld Using Integrated Acoustic Emission and Digital Image Techniques

**Ronghua Zhu [1,*] and Dazhao Chi [2]**

[1] College of Locomotive and Vehicle, Nanjing Vocational Institute of Railway Technology, Nanjing 210031, China

[2] State Key Laboratory of Advanced Welding and Joining, Harbin Institute of Technology, Harbin 150001, China; dzchi@hit.edu.cn

\* Correspondence: zhuronghua@njrts.edu.cn

**Abstract:** The three-point bending damage process of an A7N01 aluminum alloy body material and weld seam in an electric multiple unit train was monitored using acoustic emission (AE) and digital image technology. The AE signal characteristics of static load damage to the aluminum alloy and weld seam were studied using the AE signal parameter and time–frequency analysis. Based on the observation of the microstructure and fracture morphology, the source mechanism of AE signals was analyzed. The experimental results indicate that AE energy, centroid frequency, and peak frequency are effective indices for predicting the initiation of cracks in A7N01 aluminum alloy and weld seams. The digital image monitoring results of the notch tip damage evolution of aluminum alloy samples confirmed the predictions based on AE energy, centroid frequency, and peak frequency for crack initiation. The AE signal source mechanism revealed that the differences in AE characteristics between the base material and weld seam can be attributed to microstructure variations and fracture modes. In summary, the AE technique is more sensitive to changes in the fracture mode and can be utilized to monitor the damage evolution of welded structures.

**Keywords:** aluminum alloy; acoustic emission; damage evolution; time–frequency analysis; AE source mechanisms

## 1. Introduction

Pursuing lighter and more robust products is the driving force behind the continuous improvement of aluminum alloys. The traditional 7000-series Al-Zn-Mg alloys find widespread use in aerospace and transportation due to their excellent mechanical properties, particularly their outstanding specific strength [1,2]. However, the welded structures of Al-Zn-Mg alloys often face challenges such as fatigue, high temperature, corrosion, and other harsh working conditions during use. Crack initiation and propagation are among the primary reasons for the ultimate failure of these welded structures [3]. Consequently, predicting structural damage and quantifying structural integrity are crucial elements for ensuring the safe and reliable operation of welded structures.

Various traditional non-destructive testing techniques, including visual testing, ultrasonic testing, eddy current testing, and X-ray inspection, are employed to guarantee the quality of welded structures [4,5]. In ultrasonic testing, a probe emits ultrasonic waves and receives reflected echoes after the sample interacts with the waves. Due to the acoustic impedance characteristics of materials, minor defects in materials, such as voids or short cracks, may alter the propagation characteristics of ultrasonic waves. Ultrasound is thus effective in capturing minute details and detecting slight changes in materials at the early stage of structural damage. However, ultrasonic testing faces challenges in detecting complex-shaped workpieces. Based on the principle of electromagnetic induction, eddy current testing relies on defects in materials that can affect eddy current patterns, thereby allowing for the prediction and evaluation of structural damage. However, eddy current

testing is limited by the penetration depth of the eddy current and can only be used to detect surface and near-surface defects.

AE technology stands out as one of the most critical non-destructive testing methods for the real-time monitoring of material or structural states. This technology utilizes the elastic energy released during crack propagation events, which propagates through materials in the form of stress waves, causing mechanical vibrations on the surface of materials. A piezoelectric sensor converts these mechanical vibrations into electrical signals, filtered and amplified to produce AE signals. Through the processing and analysis of AE signals, the state of materials or structures can be accurately assessed [6]. Notably, the AE technique is not sensitive to the geometry of workpieces and can be used to detect components and parts with complex shapes when combined with an appropriate sensor group.

Reviewing the literature indicates that several scholars have concentrated on the correlation between AE parameters (energy, amplitude, count, rise time, etc.) and material behavior [7–9]. Ali et al. revealed the relationship between the size of corrosion and the AE parameters to detect the valve faults in the reciprocating compressor [10]. Šofer et al. [11] studied the distinction between delamination/debonding failure modes for the carbon-fiber-reinforced polymer composite tubes by using the difference in the AE energy as well as the amplitude values of both mechanisms. D'Angel and Ercolino [12] showed that AE entropy can be considered as a fracture-sensitive phenomenon for the real-time assessment of metal plates under fatigue loading. Sagasta et al. [13] investigated the correlation of AE parameters (energy b-value, bE) with local damage evolution of the reinforced concrete frame. It was found that bE values lower than 1 establish the onset of severe concrete cracking (corroborated by the visual observation of the first macro-cracks) in the vicinity of the connections. Wu et al. [14] studied the AE behavior (AE count and energy) of pitting corrosion on a vertically positioned 304 SS specimen. The potential of the in-situ evaluation of the pitting mechanism by monitoring the AE energy was verified. Moreover, Hou et al. [15] investigated the correlation between the characteristics of an AE signal (for example, count, energy, and amplitude) and the damage modes of a needled C/SiC composite during fatigue tests. Maleki et al. [16] utilized AE energy to characterize the failure of specimens and discriminated between AE signals from aluminum cracking and adhesive layer failure according to their energy content. Qiu et al. [17] combined wavelet transform and conventional AE parameter analysis to study the damage evolution behavior of asphalt mixtures. These AE characteristics are only related to the captured signals and are independent of the source of the signal and wave propagation. Accordingly, the loss of some useful information from AE sources is inevitable.

Waveform analysis is a method that can acquire information regarding AE sources based on the time-domain waveforms of AE signals and corresponding spectra recorded by the AE system. In waveform analysis, the waveforms of AE digital signals are recorded and analyzed; thus, this technique can be considered as an alternative to parametric analysis. In this regard, Zhang et al. [18] studied the stress corrosion cracking (SCC) of 304 stainless steel in high-temperature water using AE waveforms. Cumulative hits of the burst and continuous AE waveforms generated from residual ligament tearing and plastic deformation, respectively, can correspond well with the SCC modes and crack growth rate. It is reliable to use the AE waveform to monitor SCC in-situ in high-temperature water. Das et al. [19] studied the automated probabilistic classification of the cracks in cementitious components using the density-dictated unsupervised clustering algorithm based on the AE waveform parameters (RA values and average frequency). The proposed approach shows promise for the prediction of the damage state in structures based on unlabeled data obtained in the field. Kong et al. [20] used frequency–amplitude characteristics to aid in understanding the sandstone deformation and fracture process after different temperature treatments, from the viewpoint of the micro–meso–macro level of sandstone materials and under loading. Furthermore, Baker et al. [21] studied the initiation and propagation of cracks in carbon fiber-reinforced toughened epoxy polymer composite laminates using

modal AE and waveform energies coupled with peak frequency data and correlated this to matrix crack density in the transverse direction. Dossary et al. [22] considered the inflection point of the decline in the AE amplitude b-value [23,24] as a precursor to detecting rupture in asphalt mixtures and determining the correlation between the duration of transient AE bursts associated with seeded defects and the actual geometry of the defect in the test rig. These results show that transient AE analysis can be applied as a powerful and robust scheme to discriminate the damage mechanisms based on full waveform analysis.

In the transportation industry, Al-Zn-Mg alloys are often rolled into sheets or forged into complex shapes, resulting in anisotropic textures and grain geometries with a high aspect ratio. In welded structures, the weld seams predominantly exhibit dendritic cast structures. Different microstructures lead to distinct AE behaviors. In this study, AE signals and crack evolution during the three-point bending damage process of the A7N01 aluminum alloy and weld seam in an electric multiple unit (EMU) train body were monitored using the AE technique and digital image technology. The AE characteristics of crack initiation and instability propagation in the A7N01 aluminum alloy were analyzed through AE parameters and waveform analyses. Our study provides a basis for predicting the damage evolution behaviors of the A7N01 aluminum alloy and weld seams.

## 2. Materials and Methods

### 2.1. Materials and Specimen Preparation

The experimental material was A7N01 aluminum alloy, widely used in high-speed trains. The A7N01 aluminum alloy plate was processed along the rolling direction by wire electrical discharge machining, and the material composition was tested by an EDM spectrometer, as shown in Table 1.

**Table 1.** Chemical composition of the A7N01 aluminum alloy.

| Material | Si | Fe | Cu | Mn | Mg | Cr | Ni | Zn | Ti | V | Zr | Al |
|---|---|---|---|---|---|---|---|---|---|---|---|---|
| A7N01 | <0.001 | 0.087 | 0.032 | 0.377 | 1.049 | 0.089 | <0.001 | 4.23 | 0.029 | 0.014 | 0.083 | Bal. |

The welding was carried out on a TIG welding machine. The welding parameters are shown in Table 2. The specimens measured 250 mm in length, 12.5 mm in thickness, and 40 mm in width, featuring a 0.2 mm $\times$ 2.5 mm slot at the bottom center. Table 3 shows the different specimens used for testing. The notch was introduced to enhance the stress concentration factor, ensuring the initiation and propagation of cracks at the notch ends.

**Table 2.** Welding parameters.

| Weld Pass | Current (A) | Voltage (V) | Welding Speed (mm/s) | Gas Flow (L/min) |
|---|---|---|---|---|
| 1st | 250 | 24 | 50 | |
| 2nd | 270 | 24 | 42 | |
| 3rd | 270 | 24 | 42 | 17 |
| 4th | 270 | 24 | 42 | |

**Table 3.** Several types of A7N01 aluminum alloy specimens.

| Types of Specimens | | Position of the Notch |
|---|---|---|
| Type A |  | Notch in base metal |
| Type B |  | Notch in weld seam |

### 2.2. Experimental Set-Up

The three-point bending tests were carried out through an electronic universal testing machine. A digital image monitoring system was used to capture the images of the notch tip regions during the tests, as shown in Figure 1. The resolution of the optical microscope was about 0.79 μm at a working distance of 6.5 mm.

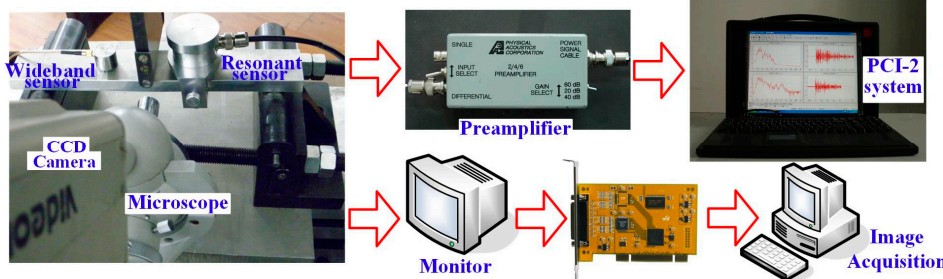

**Figure 1.** Configuration of the test set-up for fatigue damage monitoring.

The AE system was used to record AE signals throughout the tests. Two AE sensors, comprising a wide-band WD-type sensor (0.1–1 MHz) and a resonant R15I-type sensor (150 kHz resonance) with an integral 40 dB preamplification, were positioned 80 mm apart and securely affixed to the specimen using Vaseline as a coupling agent, as depicted in Figure 2. The signals from the sensors (two in total) were amplified by 40dB, digitized at 5 MHz, and stored in the AE analysis system.

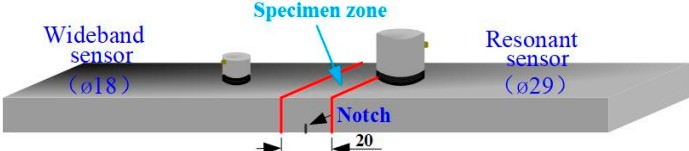

**Figure 2.** Sample size and sensor arrangement.

The A7N01 aluminum alloy sample, equipped with an AE sensor, was placed on the loading platform of the universal material testing machine, with the three-point bending loading span set to 200 mm. Cables connected the AE sensor to the preamplifier and the PCI-2 AE system host. Before the test, a preload of 100 N was applied to the sample, and colloidal sound insulation material was inserted between the indenter and the sample to reduce friction noise. The experimental parameters for AE monitoring during the static load damage of the aluminum alloy are shown in Table 4. Subsequently, the universal material testing machine was started, triggering the simultaneous collection of AE signals and notch tip damage images. The sample was loaded until failure, and the collection of AE signals and notch tip damage images was terminated simultaneously.

### 2.3. Data Filtering Procedures

In applying the AE sensor during the three-point tests, the AE monitoring system records genuine hits from crack growth and noise. One type of AE hit is induced by electromagnetic interference (EMI). Most EMI noise was identified with a maximum amplitude below 26 dB. Therefore, this threshold (40 dB) was used to minimize the EMI noise. Another type of AE hit is caused by friction between the specimen, loading roller pin, and supporting roller pins. Plastic shims between the specimen and the loading roller pin were used to minimize the noise caused by friction. Following the linear location technique, the source was positioned at the notch tip, where the time-of-arrival difference for the wave at the two sensors is zero. When the sources were positioned at the contact surface between the specimen and the two supporting roller pins, the time-of-arrival difference for the wave at the two sensors was 16.9 μs (with a wave velocity of 4725 m/s). The specimen zone (as shown in Figure 2) was analyzed using AE in order to screen out any hits influenced by friction noise. The most widely used signal measurement parameters are counts, amplitude,

duration, rise time and the measured area under the rectified signal envelope (MARSE, or alternatively counts energy) as illustrated in Figure 3.

**Table 4.** Experimental parameters for damage monitoring of A7N01 aluminum alloys.

| Specimen Number | | Load Rate (mm/min) | Notch Length (mm) | Peak Loads (KN) | Deflection (mm) | AE Activity | | Crack-Related AE Activity |
|---|---|---|---|---|---|---|---|---|
| | | | | | | Sensor 1 | Sensor 2 | |
| Base metal | 1 | 1.2 | 2.5 | 8.869 | 10.7 | 8631 | 3665 | 528 |
| | 2 | | | 8.881 | 11.3 | 7125 | 3113 | 439 |
| | 3 | | | 9.021 | 10.8 | 6565 | 3285 | 408 |
| | 4 | | | 8.956 | 6.1 | 1119 | 315 | 223 |
| | 5 | 2.4 | | 9.053 | 7 | 1895 | 1775 | 103 |
| | 6 | | | 8.972 | 6.9 | 1976 | 1634 | 105 |
| | 7 | 3.6 | | 8.910 | 7.1 | 2460 | 1749 | 93 |
| | 8 | | | 8.985 | 7.1 | 2108 | 2114 | 86 |
| | 9 | 1.2 | 1.25 | 11.546 | 11.2 | 1507 | 2125 | 365 |
| | 10 | | | 11.420 | 10.8 | 1154 | 2116 | 223 |
| | 11 | | | 11.389 | 10.7 | 1038 | 1700 | 194 |
| | 12 | 1.2 | 3.75 | 7.140 | 6.4 | 2314 | 1956 | 154 |
| | 13 | | | 7.192 | 6.4 | 2244 | 2217 | 161 |
| | 14 | | | 7.170 | 6.1 | 2424 | 2028 | 168 |
| Weld | 1 | 1.2 | 2.5 | 6.447 | 16.9 | 634 | 385 | 228 |
| | 2 | | | 6.180 | 16.1 | 546 | 306 | 182 |
| | 3 | | | 6.096 | 16 | 387 | 183 | 140 |
| | 4 | | | 6.159 | 7.4 | 579 | 377 | 213 |

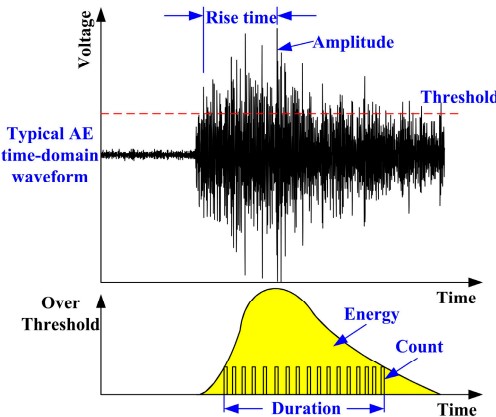

**Figure 3.** Time-domain features of AE signals.

## 3. Results and Discussion

### 3.1. Microstructural Characterization

The optical microscopy images of the microstructures for the A7N01 aluminum alloy base metal, weld and fusion zone are shown in Figure 4. The microstructure of the base metal exhibited a rolling organization comprising an $\alpha$-Al solid solution, large second-phase particles, and a small number of inclusions, as shown in Figure 4a. However, the microstructure of the weld consisted mainly of a large cast dendritic structure characterized by $\alpha$-Al solid solution, large second-phase particles, and some inclusions, as seen in Figure 4b. The welding properties diminished because the cast structure contained a few minor defects. In the fusion zone of the A7N01 welded joint near the edge of the weld (as shown in Figure 4c), coarse equiaxed crystals can be observed on one side of the weld. On the heat-affected zone side, columnar crystals predominantly align along the direction of heat dissipation adjacent to the fully recrystallized and partially recrystallized structures. The heat-affected zone of the welded joint is located within a range of approximately 15 mm from the center of the weld. The microstructure of the heat-affected zone closely resembles the optical microstructure of the base metal area. Moreover, distinct fibrous processing

traces are still visible. Thus, the AE behavior of the weld during the three-point bending tests is expected to be different from that of the base metal.

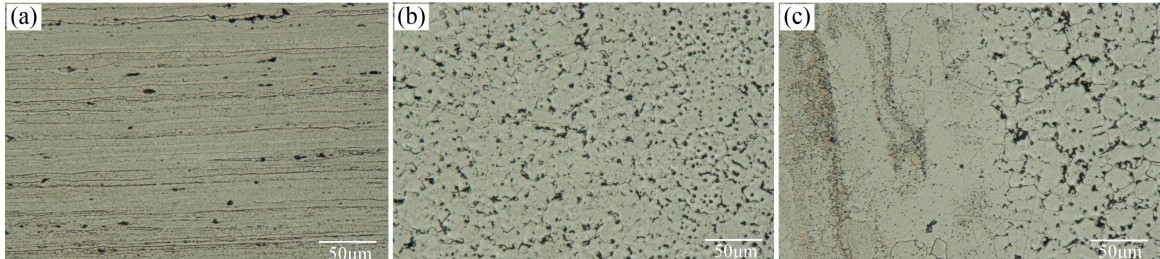

**Figure 4.** Microstructure of A7N01 aluminum: (**a**) base metal, (**b**) weld, (**c**) fusion zone.

### 3.2. Damage Evolution and AE Behavior

Figures 5 and 6 show the instantaneous AE energy, loading data, and the accumulation of AE energy during three-point bending tests on the base metal and the weld seam. According to the three peak points of the AE energy (marked as 1, 2, and 3), the entire experimental time can be divided into four parts (marked as I, II, III, and IV), as seen in Figures 5 and 6.

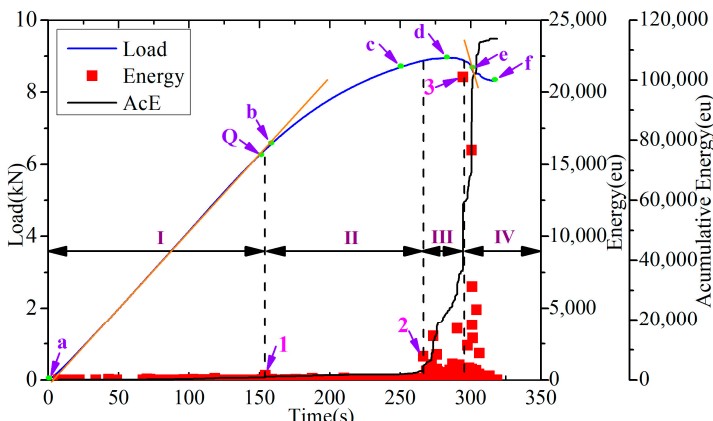

**Figure 5.** AE energy, accumulative energy, and load variations over time during the base metal test.

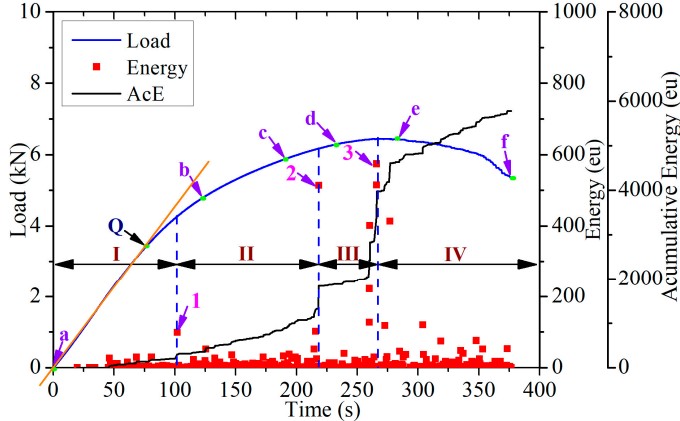

**Figure 6.** Variations in AE energy, accumulative energy, and load with respect to time during the weld seam test.

The initial AE signals indicate the onset of plastic deformation in the base metal at the notch tip, occurring at 154 s, with AE energy and load values of 319 eu and 6.4 kN, respectively, as depicted in Figure 5. However, the onset of plastic deformation in the weld occurs earlier at 102 s, when the AE energy and load are 99 eu and 4.27 kN, respectively, as

indicated in Figure 6. AE activity from the base metal steadily increases, reaching its peak at 266 s, with AE energy and load being 1626 eu and 8.88 kN, respectively. It indicates that a crack initiation event has occurred. The weld seam specimen is characterized by a more considerable increase in AE energy at 218 s, when the AE energy and load are 514 eu and 6.17 kN, respectively. It signifies the occurrence of a crack event. However, the maximum increase in AE activity from the base metal is at 295 s, when the AE energy and load are 21,051 eu and 8.9 kN, respectively. It suggests that severe damage occurred before the final failure. The AE energy of the weld remains steady until 266 s, followed by a sudden increase, signaling a new event of severe damage before eventual failure.

The digital image monitoring technique reveals that the damage process in the specimens occurs in three stages, as shown in Figures 7 and 8. In the first stage, because of stress concentration at the notch tip, slip bands are observed at 45° and 135° on the surface of the notch tip, as observed in Figures 7c and 8c. With an increase in load, the fracture and decohesion of the inclusions and secondary particles develop into microholes. The assembly process, which forms the microcrack and is responsible for microhole formation, growth, aggregation, and rupture, is depicted in Figures 7d and 8d. With the growth of microcracks, more AE activities are recorded, and the accumulative energy increases exponentially. When the crack reaches a particular size, the small cracks become large enough to overgrow. The weld specimen exhibits damage progress similar to that of the base metal, although the bending radius is larger than that of the base metal.

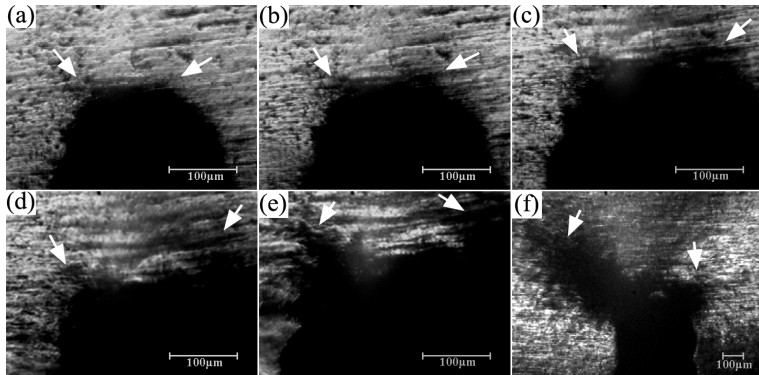

**Figure 7.** Evolution of notch tips in the base metal during three-point bending tests: (**a**–**f**) correspond to markers "a–f" in Figure 5.

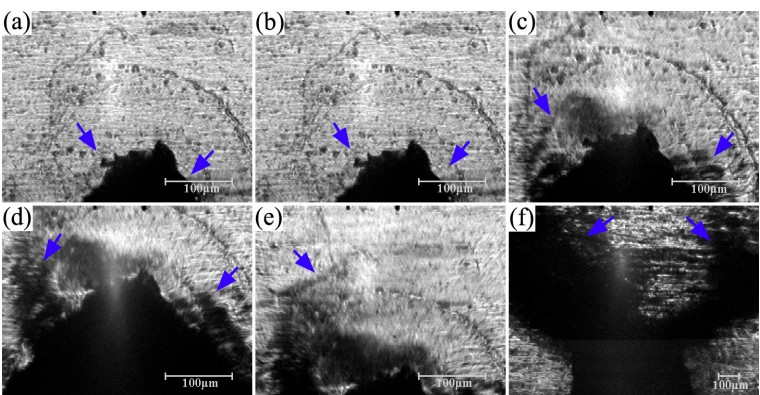

**Figure 8.** Evolution of notch tips in the weld during three-point bending tests: (**a**–**f**) correspond to markers "a–f" in Figure 6.

### 3.3. AE Waveform Characteristics

3.3.1. AE Time-Domain Characteristics

The time-domain characteristic parameters of AE signals, including amplitude, energy, count number, and rise time, were extracted using the parameter analysis method. Figure 9

shows the variation diagrams of time-domain characteristic values of AE signals from the A7N01 aluminum alloy base metal and weld seam during the three-point bending process. As observed, the characteristic values of AE signals from the A7N01 aluminum alloy base metal and the weld seam exhibit a sharp increase during the crack initiation stage. However, in the rapid crack propagation stage, the time-domain characteristic values of AE signal energy and count number for the base metal continue to rise sharply. At the same time, those for the weld seam begin to decrease and ultimately stabilize at relatively low values. Throughout the static load damage process, the characteristic value of the AE signals from the weld seam is consistently much smaller than that of the base material. Table 5 shows the statistics of characteristic values of AE signals for the base metal and weld seam during the three-point bending process.

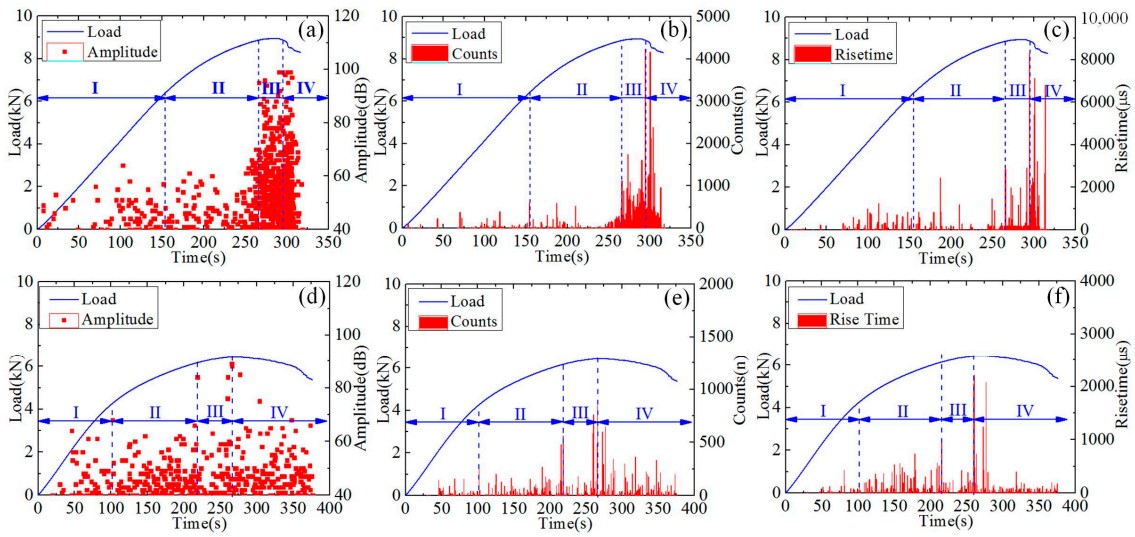

**Figure 9.** The variations in the AE characteristics of (**a**–**c**) the base metal, and (**d**–**f**) the weld.

**Table 5.** Statistics of AE characteristic parameters of the base metal and weld.

| AE Characteristics | Base Metal | | | Weld | | |
|---|---|---|---|---|---|---|
| | Variation Range | Average Value | Peak Value | Variation Range | Average Value | Peak Value |
| Amplitude (dB) | 40–99 | 53 | 99 | 40–89 | 47 | 89 |
| Energy (eu) | 0–21,051 | 102 | 21,051 | 0–547 | 13 | 547 |
| Risetime (μs) | 0–8291 | 137 | 8291 | 0–2217 | 93 | 2217 |
| Counts (n) | 0–4223 | 134 | 4223 | 1–849 | 47 | 849 |

### 3.3.2. AE Frequency-Domain Characteristics

The AE signals from the A7N01 aluminum alloy base metal and weld seam were processed via fast Fourier transform, and the centroid frequency and peak frequency were extracted as the frequency-domain characteristics, respectively.

Figure 10 shows the variation diagrams of AE signal centroid frequency and load with time for the A7N01 aluminum alloy base metal. As depicted in Figure 10a, the centroid frequencies of AE signals collected by the wideband sensor before 266 s are mainly concentrated in the frequency band range of 297~536 kHz. As the load increases, after 266 s, the AE signals gradually intensify, and the frequency band range gradually expands, featuring the appearance of some low-frequency signals in the range of 142~297 kHz among the centroid frequencies. The variation trend of the centroid frequency of AE signals collected by the resonant sensor (as shown in Figure 10b) is similar, with low-frequency signals of 133~177 kHz appearing after 266 s. Therefore, the variation in the centroid

frequency of AE signals can be used to deduce crack initiation in the A7N01 aluminum alloy base metal.

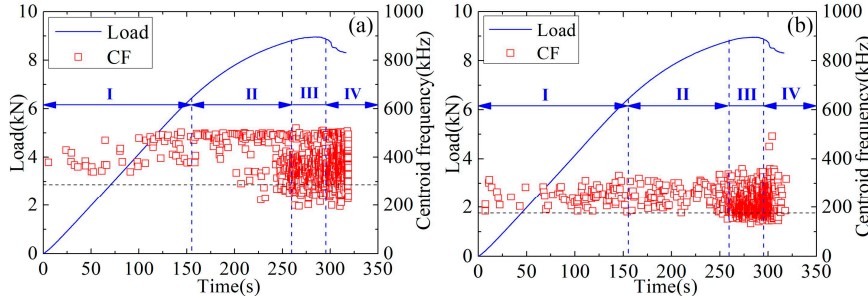

**Figure 10.** The evolution of AE centroid frequency for the base metal: signals from (**a**) wide-band sensor and (**b**) resonant sensor.

Figure 11 shows the AE centroid frequency variation diagrams and load with time for the A7N01 aluminum alloy weld seam. As shown in Figure 11a, the variation trend of the centroid frequency of AE signals from the weld seam collected by the wide-band sensor differs from that of the base material. The centroid frequency does not shift to the low-frequency range after crack initiation. In the early stage, centroid frequencies are mainly concentrated in the frequency band range of 311~515 kHz; after 147 s, high-frequency signals emerge in the frequency band range of 515~660 kHz. The change in the centroid frequency of AE signals collected by the resonant sensor mirrors that of the wide-band sensor. In the initial stage of the experiment, the centroid frequency of AE signals collected by the resonant sensor is mainly concentrated in the frequency band of 151~222 kHz, and after 147 s, the centroid frequency in the frequency band of 222~316 kHz appears. Therefore, the crack initiation in the A7N01 aluminum alloy weld seam can also be inferred from the variation in the centroid frequency of AE signals.

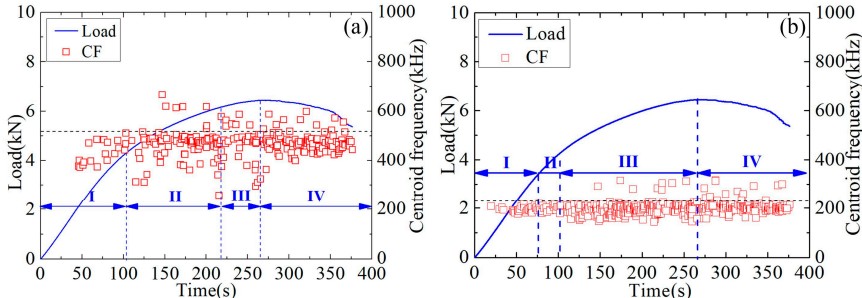

**Figure 11.** The evolution of AE centroid frequency for the weld: signals from (**a**) wide-band sensor and (**b**) resonant sensor.

Figure 12 shows the distribution of peak frequencies collected by the resonant sensor for the A7N01 aluminum alloy base metal and weld seam throughout the loading period. As illustrated in Figure 12a, the AE peak frequencies of the base metal predominantly fall within the frequency ranges of 96~107 kHz, 220~255 kHz, and 579~643 kHz throughout the entire duration, suggesting familiar sources for these AE events, such as dislocation movement during plastic deformation. However, at approximately 266 s (after stage A in Figure 12a), a high-amplitude low-frequency signal appears in the frequency range of 112~188 kHz, corresponding to the energy mutation point 2 in Figure 5. AE signals in this frequency range persist until the end of the experiment, indicating that crack initiation and propagation in the base metal generate many low-frequency AE signals. It also explains the shift in centroid frequency to the lower frequency band (below 297 kHz) after crack initiation.

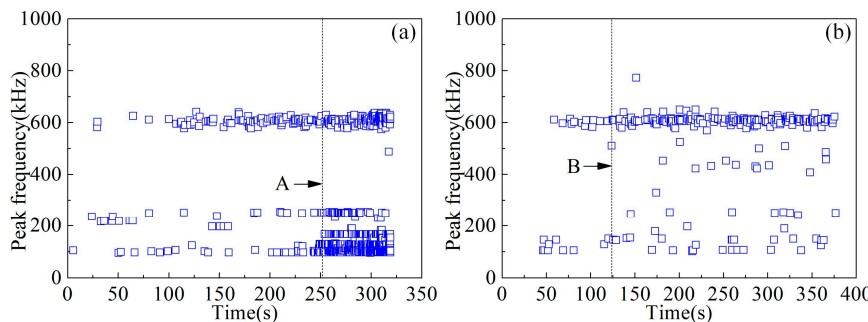

**Figure 12.** The evolution of AE peak frequency in A7N01 aluminum alloy: (**a**) the base metal, (**b**) the weld.

Compared to the base material, the peak frequencies of the weld seam are mainly distributed in the frequency ranges of 100~154 kHz and 570~650 KHZ before stage B, as shown in Figure 12b. These AE signals mainly originate from the plastic deformation of the notch tip and around the inclusions. As the loading process advances to stage B, additional peak frequencies appear in the frequency range of 242~525 kHz and persist until the end of the experiment. These peak frequencies correspond to the initiation and propagation of weld cracks. From Figure 11a, the AE centroid frequencies of the weld seam in the early stage are distributed in the frequency range of 311~515 kHz. However, in stage B, many centroid frequencies exceeding 515 kHz appear, likely due to the appearance of peak frequency signals in the frequency range of 242~525 kHz.

### 3.3.3. AE Time-Frequency Domain Characteristics

Time-frequency analysis provides joint distribution information in both the time and frequency domains, thus clearly describing the variation relationship between signal frequency and time. Based on the analysis of time-domain and frequency-domain characteristics in both the base material and weld seam, typical AE signals depicted in Figure 13 at various stages are selected. These stages correspond to mutation points 1, 2, and 3 in the energy change process of the base material and weld seam, as illustrated in Figures 5 and 6, respectively. The selected signals underwent wavelet transform analysis to examine their time-frequency variation characteristics.

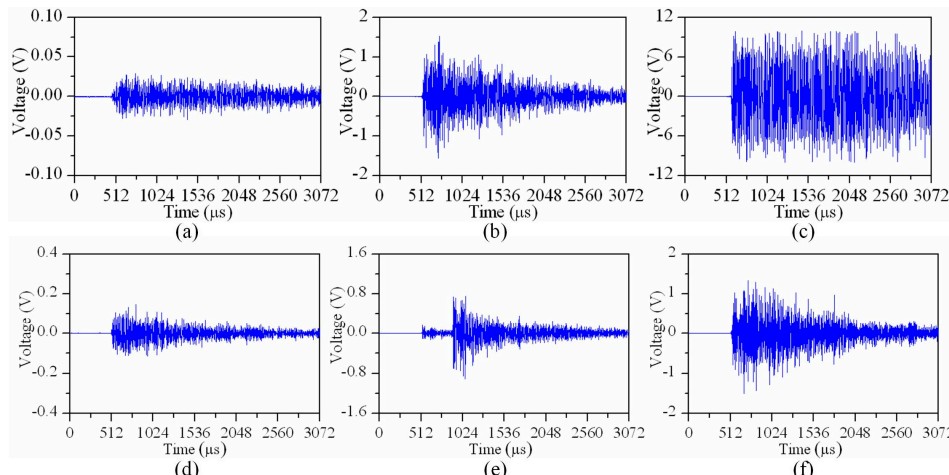

**Figure 13.** The time-domain waveforms of AE signals for (**a**–**c**) the base metal and (**d**–**f**) the weld.

Figures 13 and 14 present the typical time-domain signal waveform diagrams during the three-point bending process of the base metal and weld seam, respectively. Plastic deformation is shown in Figures 13a,d and 14a,d. Similarly, crack initiation can be seen in Figures 13b,e and 14b,e. Moreover, crack propagation is given in Figures 13c,f and 14c,f.

The statistical characteristic values of the AE time-domain waveform diagrams are listed in Table 6.

**Figure 14.** The wavelet scalograms of AE signals for (**a**–**c**) the base metal, and (**d**–**f**) the weld.

**Table 6.** Eigenvalues of typical AE signals for the base metal and weld.

| Specimen Type | Waveform | Amplitude (dB) | Energy (eu) | Risetime (μs) | Duration (μs) | Counts (n) |
|---|---|---|---|---|---|---|
| Base metal | Bm-a | 54 | 319 | 28 | 5569 | 583 |
| | Bm-b | 96 | 3045 | 17 | 24,157 | 1741 |
| | Bm-c | 99 | 21,051 | 327 | 30,000 | 4223 |
| Weld | W-a | 63 | 45 | 148 | 2084 | 550 |
| | W-b | 79 | 206 | 269 | 3606 | 1009 |
| | W-a | 84 | 539 | 82 | 5028 | 1874 |

In Figure 13a,b, the waveforms of the base metal are burst waveforms with a short rise time. In contrast, the waveform is a mixed waveform of burst and continuous signals, characterized by a longer rise time and a more prolonged duration at high amplitudes in Figure 13c. This phenomenon is mainly associated with strain energy release during crack propagation. Micro-crack propagation presents a pulse attenuation wave during crack initiation and steady propagation stages. In contrast, the strain energy released during crack instability propagation is more prominent and lasts longer. In comparison with the base material, all three typical AE waveforms of the weld seam, as shown in Figure 14, exhibit burst waveforms.

### 3.4. AE Source Mechanisms

Figure 15 represents the SEM images of the notch tip region of the base metal and weld. Figure 15b,d are the magnified portions represented by the yellow rectangles in Figure 15a,c, respectively. In the base metal sample, a significant fraction of the notch tip contained elongated dimples (A), small equiaxed dimples (B), and a few secondary cracks (C), as illustrated in Figure 15a. The dimples are small and typically a few μm in diameter. Behind the area of small equiaxed dimples, de-lamination bands (D) are also found. However, the fracture surfaces of the welds show noticeable distinctions. Unlike the base metal, in the weld, an elongated dimple area (A), some secondary cracks (C), and an area of an equiaxed dimple (B) are found ahead of the notch. In this case, the dimples are more extensive, with 10–30 μm diameters. These secondary cracks appear initiated at the oxides formed during the welding process. Some gas pores (P) can also be seen in the fracture surface, which CO, hydrogen, or nitrogen may have caused.

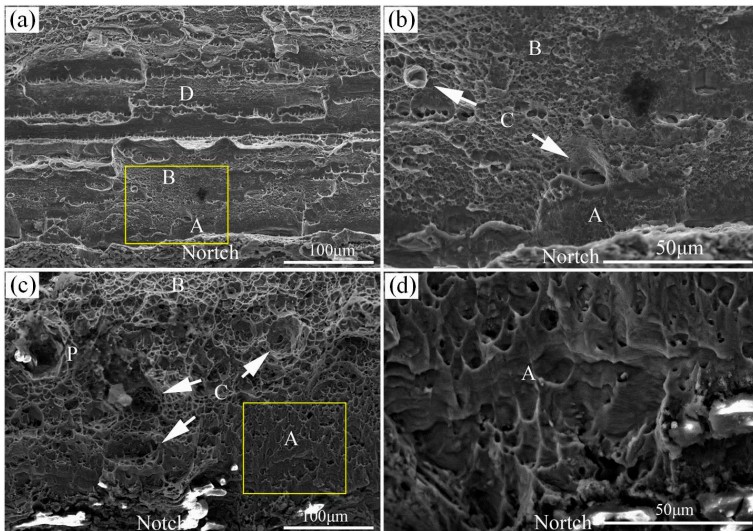

**Figure 15.** SEM images of fracture surfaces: (**a**,**b**) the base metal and (**c**,**d**) the weld.

Intense AE events can be generated by plastic deformation, crack initiation, and propagation, which contribute to the rapid growth of AE energy. Because of the stress concentration at the notch tip, elongated dimples are produced in the region, as shown in Figure 15b,d. The origin of AE during the crack initiation stage is attributed to microvoid nucleation at the boundaries of the inclusions and the second-phase particles, resulting from their brittleness and rapid fracture. Digital image monitoring (Figures 7 and 8) reveals that the microcracks are initiated midway between the yield and maximum load. The AE signals are generated mainly in the same loading range. Hence, the AE generation can be used as the indicator of crack initiation. The crack propagation occurs primarily because of the microcrack coalescence in the A7N01 aluminum alloy. This mechanism served as the primary source of the AE recorded during the rapid crack propagation stage. In contrast to the weld, the base metal shows delamination defects, which may account for the higher AE energy shown by the base metal than that by the weld during the rapid crack propagation.

### 3.5. Influencing Factors for AE Features

Various AE behaviors can manifest during material damage, with some materials generating stronger AE signals and others producing less. Materials may exhibit different AE behaviors under different load conditions. Therefore, it is necessary to study the factors influencing the AE behaviors of aluminum alloys.

Figure 16 shows the load and AE cumulative count variation diagrams for the A7N01 aluminum alloy base metal at different loading rates. In Figure 16a, it is evident that the bending strength of the base metal presents a decreasing trend with increasing loading rates. Figure 16b reveals that, under the same deflection of the base material, the cumulative count of AE gradually increases during the crack initiation and propagation stage with higher loading rates. It happens because the stress level of the notch tip is relatively large under high loading rates, inducing the release of a substantial stress wave by the base material.

Figure 17 shows the load and AE cumulative count variation diagrams for the aluminum alloy base metal under different notch lengths. Figure 17a shows that, with the increase in notch length, the bending strength of the A7N01 aluminum alloy sample gradually decreases, indicating reduced resistance to bending deformation. In Figure 17b, the peak energy of AE during the crack propagation stage increases with the notch length.

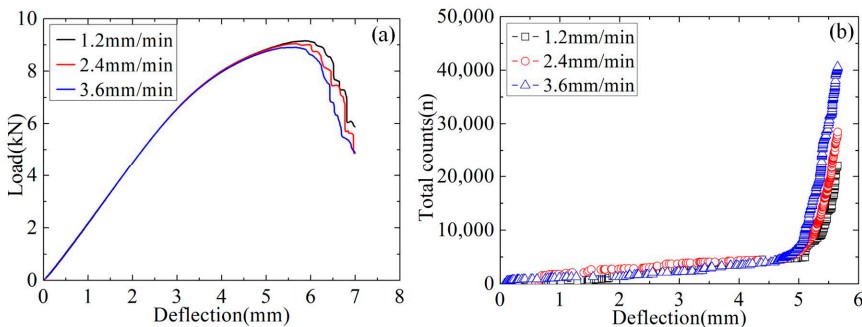

**Figure 16.** Effect of loading rate on load and AE count: (**a**) load curve, (**b**) accumulative AE count.

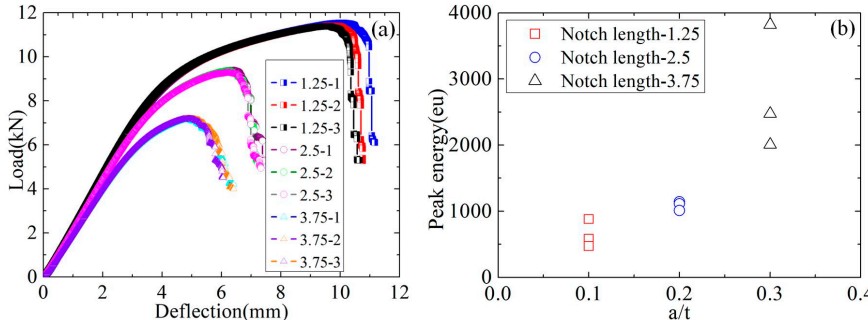

**Figure 17.** Effect of notch length on load and AE energy: (**a**) load curve, (**b**) peak AE energy.

Figure 18 shows the load and AE cumulative count variation diagrams for the aluminum alloy base metal, weld seam, and heat-affected zone during the three-point bending process. Figure 18a reveals a decreasing trend in bending strengths for the base metal, heat-affected zone, and weld seam, which are 426 MPa, 404 MPa, and 309 MPa, respectively. In the heat-affected zone, the fine second-phase particles and precipitated strengthened phase particles dissolve into the matrix under the action of welding thermal cycling, resulting in the coarsening of grain size and leading to a de-graded bending strength. As part of the cast microstructure, the weld seam contains minor defects like pores and inclusions, negatively impacting the bending strength. Figure 18b shows an increasing trend in the total AE count for the base material, heat-affected zone, and weld seam during the crack initiation and propagation stage.

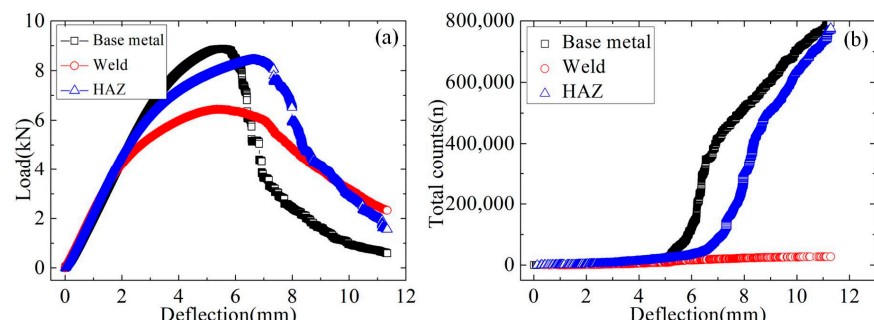

**Figure 18.** Effect of microstructure on load and AE count: (**a**) load, (**b**) total AE counts.

At the crack initiation and propagation stage, under the same deflection, the total AE count of the base material surpasses that of the heat-affected zone, and the total AE count of the base material or heat-affected zone significantly exceeds that of the weld seam. It is mainly due to the fracture microstructure of the base material closely resembling that of the heat-affected zone. The base material and heat-affected zone exhibit small dimples near the notch tip of fracture samples, with a layered structure appearing farther from the notch tip. In contrast, the fracture of the weld seam sample is essentially a dimple structure, as shown in Figure 19.

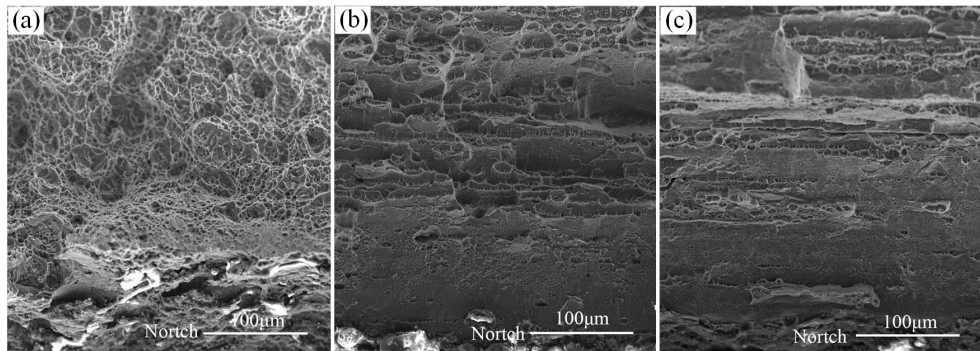

**Figure 19.** Comparison of different fracture surfaces: (**a**) weld, (**b**) HAZ, (**c**) base metal.

## 4. Conclusions

The static load damage of the A7N01 aluminum alloy base metal and weld seam samples were thoroughly studied through AE parameter analysis and waveform analysis.

(1) AE energy can be used to indicate crack initiation. Digital images obtained by monitoring the notch tip region of an A7N01 aluminum sample confirm the predictions based on AE signals. The prediction based on AE energy was validated by monitoring damage images at the notch tip of the A7N01 aluminum alloy sample.

(2) Frequency–domain characteristic analysis showed that the transformation of centroid frequencies occurred after crack initiation in both the base material and weld seam. The centroid frequencies of the base material were transformed to the low-frequency range of 297~536 kHz, while those of the weld seam shifted to the high-frequency range of 515~660 kHz. Moreover, after crack initiation, new peak frequencies appeared in the frequency range of 112~188 kHz and 242~525 kHz for the base material and weld seam, respectively. Therefore, centroid and peak frequencies can also indicate crack initiation in aluminum alloys.

(3) SEM images of the fractured surfaces indicate that the base metal exhibits smaller dimples and fewer de-lamination defects than the weld. It might be the main reason for the low energy emission, weak signal strength, and low peak amplitude observed in the welded specimens. In contrast, the base metal specimens exhibit more substantial AE energy, higher amplitude, and greater AE event counts.

(4) With the increase in loading rate and notch length, the AE cumulative count of the base metal presented a gradually increasing trend during both the crack initiation and propagation stages. The total AE counts of the base material, heat-affected zone, and weld seam also increased during crack initiation and propagation.

**Author Contributions:** Conceptualization, R.Z.; methodology, R.Z.; software, D.C.; validation, R.Z.; formal analysis, D.C.; investigation, R.Z.; resources, R.Z.; data curation, R.Z.; writing—original draft preparation, R.Z.; writing—review and editing, R.Z.; visualization, D.C.; supervision, D.C.; project administration, D.C.; funding acquisition, R.Z. and D.C. All authors have read and agreed to the published version of the manuscript.

**Funding:** This research was sponsored by Natural Science Foundation of China (Grant No. 52375328) and was supported by CGN-HIT Advanced Nuclear and New Energy Research Institute (Grant No. CGN-HIT202310), High-Speed Rail Safety Collaborative Innovation Center for Chinese Ministry of Education (Grant No. GTAQ2021007), the General Project of Natural Science Research in Colleges of Jiangsu Province (Grant No. 21KJB580018), and the Rail Transit Equipment Digital Research Center (Grant No. KQPT202301).

**Institutional Review Board Statement:** Not applicable.

**Informed Consent Statement:** Not applicable.

**Data Availability Statement:** The data presented in this study are available on request from the corresponding author.

**Conflicts of Interest:** The authors declare no conflicts of interest.

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
