# Peer review of "Failure Characterization of Al-Zn-Mg Alloy and Its Weld Using Integrated Acoustic Emission and Digital Image Techniques"

_metals, doi:10.3390/met14020190_

Round 1

Reviewer 1 Report

Comments and Suggestions for Authors

1) The amount of literature reviewed is insufficient. Please consider more recent literature that is related to welding and microstructure inspection work.

2) Images of installations are best shown schematically.

3) The exact composition of the alloy will help readers. It is necessary to add the composition and method for its determination.

4) “Various traditional non-destructive testing techniques, including visual testing, ul-37 trasonic testing, eddy current testing, and X-ray inspection, are employed to guarantee 38 the quality of welded structures [4-5].” As far as I understand, your setup is capable of picking up a signal during a dynamic test. How are you going to perform non-destructive testing to replace existing methods?

Author Response

Dear Reviewer:

Thank you for your comments concerning our manuscript entitled “Failure Characterization of Al-Zn-Mg alloy and its weld using integrated acoustic emission and digital image techniques” (ID: metals- 2796968). Those comments are all valuable and very helpful for revising and improving our paper, as well as the important guiding significance to our researches. We have studied comments carefully and have made correction, which we hope meet with approval. Revised portion are marked in red in the paper. The main corrections in the paper and the responds to the reviewer’s comments are listing bellowing.

We tried our best to improve the manuscript and made some changes in the manuscript. These changes will not influence the content and framework of the paper. And here we did not list the changes but marked in red in revised paper.

We appreciate for reviewer warm work earnestly, and hope that the correction will meet with approval.

Once again, thank you very much for your comments and suggestions.

Best regards,

Ronghua zhu et al.

Reviewer 2 Report

Comments and Suggestions for Authors

The paper deals with the "Failure Characterization of Al-Zn-Mg alloy and its weld using integrated acoustic emission and digital image techniques". The results are interesting, but the following points should be considered before publication:

1. Please discuss the locations of crack initiation.

2. Please indicate the grain sizes of the different parts of the sample.

3. What influence do the grain sizes have on the properties and fracture behaviour?

4. Please add IPF and KAM maps with EBSD.

5. In the introduction, please add some information about the defects that can act as crack initiation sites. For example, this paper states that second phase particles can cause cracking.

https://doi.org/10.1016/j.jallcom.2020.157300

Comments on the Quality of English Language

Moderate editing of English language required.

Author Response

(The authors gave the same response as above.)

Reviewer 3 Report

Comments and Suggestions for Authors

The text appears to be well-written, with no visible substantive, linguistic or grammatical errors.

The sentences are clear, and the information is presented in a concise and structured manner. There is a consistency in terminology and usage throughout the text. However, there are a few suggestions for clarification and consistency:

However, there are some suggestions for clarification and consistency of text in the conclusions section:

    In (1), the sentence "The digital images obtained by monitoring the notch-tip region of a A7N01 aluminum sample..." could be revised for smoother flow:

    Revised: "Digital images obtained by monitoring the notch-tip region of an A7N01 aluminum sample confirm the predictions based on AE signals."

    In (3), consider rephrasing the sentence for clarity:

    Revised: "This may be the main reason for the emission of low energy, weak signal strength, and low peak amplitude from the welded specimens, and stronger AE energy, amplitude, and more AE event counts observed in the base metal specimens."

    In (4), consider clarifying the phrase "during the crack initiation and propagation stage" for smoother expression:

    Revised: "With the increase in loading rate and notch length, the AE cumulative count of the base metal presented a gradual increasing trend during both the crack initiation and propagation stages."

Overall, these are minor adjustments to enhance the coherence and flow of the text.

Author Response

(The authors gave the same response as above.)

Reviewer 4 Report

Comments and Suggestions for Authors

Describe how the specimens were manufactured and how the notch was incorporated

Provide more and enough information of the equipment used (3 point bending, digital monitoring, AE, ...) which would allow the reproduction of the results presented in the manuscript.

Comments on the Quality of English Language

Revise the English grammar and also adjust properly the past and present tense along the manuscript because it is missed up in most part

Author Response

(The authors gave the same response as above.)

Round 2

Reviewer 1 Report

Comments and Suggestions for Authors

Accept in present form

Author Response

Dear Reviewer:

Thank you for your comments concerning our revised manuscript entitled “Failure Characterization of Al-Zn-Mg alloy and its weld using integrated acoustic emission and digital image techniques” (ID: metals- 2796968). 

Best regards,

Ronghua zhu et al.

Reviewer 2 Report

Comments and Suggestions for Authors

The comments should be done point by point.

Comments on the Quality of English Language

Extensive editing of English language required.

Author Response

Dear Reviewer:

Thank you for your comments concerning our revised manuscript entitled “Failure Characterization of Al-Zn-Mg alloy and its weld using integrated acoustic emission and digital image techniques” (ID: metals- 2796968). Those comments are all valuable and very helpful for revising and improving our paper, as well as the important guiding significance to our researches. We have studied comments carefully and have made extensive editing of English language, which we hope meet with approval.

In this paper, we focus on the AE characteristics of the A7N01 aluminum alloy base metal and weld during static load damage, and find out the indicators of crack initiation for the A7N01 aluminum alloy. The microscopic analysis of crack initiation and propagation of the A7N01 aluminum alloy is not the focus of this paper, and will be carried out in detail in subsequent studies (for example, EBSD analyses).

We appreciate for reviewer warm work earnestly, and hope that the correction will meet with approval.

Once again, thank you very much for your comments and suggestions.

Best regards,

Ronghua zhu et al.

Round 3

Reviewer 2 Report

Comments and Suggestions for Authors Almost all comments have been postponed to the next study. I regret to inform you that I cannot accept this work in its present form. In this state, the manuscript is incomplete. If there is no access to the facilities, the author can use the literature to complete their discussion. Comments on the Quality of English Language

-